# Sociodemographically differential patterns of chronic pain progression revealed by analyzing the all of us research program data

Edwin Baldwin[1], Jin Zhou[2], Wenting Luo[3], W. Michael Hooten[4]*, Jungwei W. Fan[5]*, Haiquan Li[1]*

**1** Department of Biosystems Engineering, University of Arizona, Tucson, Arizona, United States of America, **2** Department of Biostatistics, University of California at Los Angeles, Los Angeles, California, United States of America, **3** Statistics and Data Science Graduate Interdisciplinary Program, University of Arizona, Tucson, Arizona United States of America, **4** Division of Pain Medicine, Department of Anesthesiology, Mayo Clinic, Rochester, Minnesota, United States of America, **5** Department of Artificial Intelligence & Informatics, Mayo Clinic, Rochester, Minnesota, United States of America

* haiquan@arizona.edu (HL); Fan.Jung-wei@mayo.edu (JWF); Hooten.William@mayo.edu (WMH)

## Abstract

The differential progression of ten chronic overlapping pain conditions (COPC) and four comorbid mental disorders across demographic groups have rarely been reported in the literature. To fill in this gap, we conducted retrospective cohort analyses using All of Us Research Program data from 1970 to 2023. Separate cohorts were created to assess the differential patterns across sex, race, and ethnicity. Logistic regression models, controlling for demographic variables and household income level, were employed to identify significant sociodemographic factors associated with the differential progression from one COPC or mental condition to another. Among the 139 frequent disease pairs, we identified group-specific patterns in 15 progression pathways. Black or African Americans with a COPC condition had a significantly increased association in progression to other COPCs (CLBP->IBS, CLBP->MHA, or IBS->MHA, OR≥1.25, adj.p ≤ 4.0x10⁻³) or mental disorders (CLBP->anxiety, CLBP->depression, MHA->anxiety, MHA->depression, OR≥1.25, adj.p ≤ 1.9x10⁻²) after developing a COPC. Females had an increased likelihood of chronic low back pain after anxiety and depression (OR≥1.12, adj.p ≤ 1.5x10⁻²). Additionally, the lowest income bracket was associated with an increased risk of developing another COPC from a COPC (CLBP->MHA, IBS->MHA, MHA->CLBP, or MHA->IBS, OR≥1.44, adj.p ≤ 2.6x10⁻²) or from a mental disorder (depression->MHA, depression->CLBP, anxiety->CLBP, or anxiety->IBS, OR≥1.50, adj.p ≤ 2.0x10⁻²), as well as developing a mental disorder after a COPC (CLBP->depression, CBLP->anxiety, MHA->anxiety, OR≥1.37,adj.p ≤ 1.6x10⁻²). To our knowledge, this is the first study that unveils the sociodemographic influence on COPC progression. These findings suggest the importance of considering sociodemographic factors to achieve optimal prognostication and preemptive management of COPCs.

which permits unrestricted use, distribution, and reproduction in any medium, provided the original author and source are credited.

**Data availability statement:** The original data can be accessed from All of Us Research Workbench, after registered as a control tier user from https://www.researchallofus.org/register/.

**Funding:** This work was supported by the National Institute of Dental and Craniofacial Research (grant 1R21DE031424 to EB, JZ, WH, JF, and HL). The funder had no role in study design, data collection and analysis, decision to publish, or preparation of the manuscript.

**Competing interests:** The authors have declared that no competing interests exist.

## Author summary

Understanding the differential progression of chronic overlapping pain conditions (COPC) can inform patients and clinicians in optimizing the disease management. Although differential manifestations have been extensively reported for individual COPCs, differential patterns along the progression from one COPC to another have been rarely studied. The study addresses this gap specifically by identifying differential COPC progression associated with sex, race, and ethnicity, providing valuable insights for patients and clinicians to plan for their prevention or treatment strategies in the context of sociodemographic relative risk.

## Introduction

Chronic overlapping pain conditions (COPCs), defined as a subset of comorbid chronic pains that co-occur at higher-than-expected levels in the general population, are driven by shared mechanisms [1] and tend to be more common in females [1]. The National Institutes of Health Pain Consortium currently recognizes ten COPCs: chronic low back pains (CLBP), migraine (MHA), irritable bowel syndrome (IBS), temporomandibular disorders (TMD), fibromyalgia (FM), chronic fatigue syndrome (CFS), chronic tension-type headache (CTTH), endometriosis (ENDO), vulvodynia (VVD), and urologic chronic pelvic pain syndrome (UCPPS). COPCs significantly affect individuals worldwide, interfering with daily activities, mental health, and overall quality of life. Given their frequent comorbidities with mental health conditions [2], our study includes the four most frequently studied conditions—anxiety, depression, bipolar disorder, and schizophrenia [2–4].

The experience of pain is not uniform but intricately shaped by the interplay of sociodemographic factors, including sex, gender, age, race, ethnicity, and financial status (e.g., income) [5,6], resulting in disparities known to particularly affect subpopulations [7]. Previous research reported a strong association between demographic background [5] and pain-related health outcomes [8], which implied unique pain mechanisms across racial/ethnic groups [9–11] and sexes [12–14]. Besides recognized biological differences, social determinants are shown to affect the experience of chronic pains [15]. Evidence indicates that people with financial challenges encounter greater healthcare barriers in accessing pain management [16]. Further, in urban populations, the intersection of sociodemographic factors has been shown to create unique identities that impact pain experiences [5,17].

Although extensive research has explored disparities in individual COPC conditions, few studies have investigated sociodemographic disparities in the incremental progression among COPCs and comorbid mental disorders. To address this gap, we examined the sociodemographic disparities in COPCs using a large-scale longitudinal cohort from the All of Us Research Program (AoURP) [18], which has enrolled participants from a broad variety of populations.

The primary objective of the research is to discover statistically significant sociodemographic factors associated with one of the ten COPC conditions and four mental health conditions. We sought to reveal the differential progression patterns among these distinct COPC and mental conditions that were known for their frequent comorbidities with COPCs [4,19]. Validation of such differential disease manifestations will allow for more precise prevention strategies that target the relative risk of further COPC complications in specific subpopulations. The study represents the first of its kind by applying rigorous statistics to identify health disparities and financial predictors in COPCs, specifically based on progression patterns. Our results contribute to understanding the disparate, nuanced clinical experiences in COPCs, advocating for both accessible and individualized chronic pain care that would fit each patient's unique needs [16,20].

## Results

Of the 309,157 participants with valid electronic health records, 88,496 (28.6%) had at least one COPC condition. Table 1 shows the demographic characteristics of the COPC cases and their comorbid mental disorders in the AoURP version 8 dataset.

Table 2 demonstrates the elevated risk of specific COPCs in females and racial minorities, based on multiple logistic regression studies of these variables using demographically matched cohorts. Significance is defined as a p-value below the 0.05 threshold after the Bonferroni correction, adjusted for other demographic factors and income levels. Except for UCPPS, all COPCs are more likely to develop in females. For instance, fibromyalgia had an odds ratio of 9.99 in females (95% CI [8.24, 12.10], adj.p $< 10^{-16}$), indicating that females are about ten times more likely to be diagnosed with fibromyalgia than males. Similarly, for migraine headaches (MHA), females exhibit an odds ratio of 3.31 (95% CI [3.09, 3.55], adj.p $< 10^{-16}$), suggesting females have more than triple the risk to have MHA compared to males. Along the race variable, Black or African American persons have an odds ratio of 1.24 (95% CI [1.16, 1.34], adj.p $< 10^{-16}$) in developing chronic lower back pain (CLBP) even after adjusting for income levels and other demographic factors (e.g., sex and age).

The onset of mental disorders also showed significant demographic associations. Females present higher odds ratios of developing anxiety and depression at 1.50 (95% CI [1.44, 1.56]) and 1.48 (95% CI [1.42, 1.54]) with adjusted p-values less than $10^{-16}$. For schizophrenia, a significant odds ratio of 2.61 (95%CI [1.93, 3.52], adj.p $< 10^{-16}$) was observed in Black or African American persons, which corroborates the literature [21,22].

Fig 1 illustrates how sex and race are associated with the development of a second COPC or mental condition following a previous condition. This network was derived from 139 frequent co-occurring (threshold: at least 21 occurrences) pairs of conditions, where each pair of nodes must involve at least one COPC, i.e., two COPCs or one COPC and one mental condition (S1 File). Each arrow between two nodes indicates the two conditions are associated sequentially using regression analyses with non-Hispanic White males as the reference group. Black or African American persons show the strongest relative risk for COPC and mental disorder progression compared to White persons (blue lines).

First, Black or African American persons are more likely to develop another COPC after the first one: those with CLBP are more likely to develop subsequent IBS (OR=1.49, 95% CI [1.07, 2.08], adj.p $= 1.2 \times 10^{-3}$) and MHA (OR=1.25, 95% CI [1.03, 1.53], adj.p $= 4.0 \times 10^{-3}$), and those with IBS are more likely to develop MHA (OR=1.69, 95% CI [1.02, 2.80], adj.p $= 1.4 \times 10^{-2}$). Second, they are more likely to develop FM after depression (OR=1.67, 95% CI [1.20, 2.32], adj.p $= 2.1 \times 10^{-6}$). Third, they are more likely to develop a mental disorder after the onset of a COPC: anxiety (OR=1.41, 95% CI [1.24, 1.61], adj.p $< 10^{-16}$) and depression (OR=1.32, 95% CI [1.15, 1.51], adj.p $= 1.2 \times 10^{-11}$) following CLBP, and anxiety (OR=1.37, 95% CI [1.11, 1.68], adj.p $= 5.5 \times 10^{-6}$) and depression (OR=1.25, 95% CI [1.00, 1.55], p $= 1.9 \times 10^{-2}$) following migraine. Of note, half of the results heightened complication risk of both other COPCs (e.g., MHA and FM) and mental disorders (e.g., anxiety and depression) added to the known enrichment of CLBP in Black or African American persons (OR=1.24, 95% CI [1.16, 1.34], adj.p $< 10^{-16}$, Table 2), among others.

In addition, we observed a sex disparity in the progression of COPC comorbidities: females were more likely to develop CLBP after depression (OR=1.15, 95% CI [1.04, 1.27], adj.p $= 6.6 \times 10^{-5}$) or anxiety (OR=1.12, 95% CI [1.00, 1.24],

**Table 1. Demographic Characteristics of the Study COPC Cases.**

|  | All of Us Research Program (n=88,496) |
|---|---|
| **Sex** |  |
| Female | 62,265 (70.4%) |
| Male | 26,231 (29.6%) |
| **Race** |  |
| American Indian or Alaska Native | 934 (1.1%) |
| Asian | 1,308 (1.5%) |
| Black or African American | 14,641 (16.5%) |
| Middle Eastern or North African | 405 (0.5%) |
| More than One Population | 4,296 (4.9%) |
| Native Hawaiian or Other Pacific Islander | 63 (0.07%) |
| None of these | 996 (1.1%) |
| Unknown | 12,547 (14.2%) |
| White | 53,306 (60.2%) |
| **Ethnicity** |  |
| Hispanic | 13,323 (15.1%) |
| Non-Hispanic | 72,679 (82.1%) |
| Others | 2,494 (2.8%) |
| **COPC Onset** |  |
| Chronic fatigue syndrome (CFS) | 3,563 (4.0%) |
| Chronic lower back pain (CLBP) | 51,480 (58.2%) |
| Chronic tension-type headache (CTTH) | 764 (0.9%) |
| Endometriosis (ENDO) | 1,770 (2.0%) |
| Fibromyalgia (FM) | 6,469 (7.3%) |
| Irritable bowel syndrome (IBS) | 8,834 (10.0%) |
| Migraine (MHA) | 23,633 (27.7%) |
| Temporomandibular disorder (TMD) | 2,381 (2.7%) |
| Urologic chronic pelvic pain syndrome (UCPPS) | 947 (1.1%) |
| Vulvodynia (VVD) | 162 (0.2%) |
| **Mental Health Conditions** |  |
| Anxiety | 30,344 (34.3%) |
| Bipolar disorder | 4,221 (4.8%) |
| Depressive disorder | 29,860 (33.7%) |
| Schizophrenia | 770 (0.9%) |

*Note*: Percentages in parentheses represent the proportion among the COPC cases only, not the prevalence of the general population. Unknown Race included: I prefer not to answer, none indicated, and skip. Others Ethnicity: none of these, prefer not to answer, and skip. Intersex and transgender persons were excluded due to insufficient statistical power.

adj.p = 1.5x10$^{-2}$), and more likely to develop schizophrenia after CLBP (OR=1.59, 95% CI [1.01, 2.52], adj.p = 2.1x10$^{-2}$). Note that we also accounted for income level while examining the disparities, and the findings aligned with the literature. Specifically, we found that lower income is associated with an elevated risk of developing another subsequent COPC or mental disorder, compared with the highest income group ($200,000 annually). For example, persons with an annual income below $25,000 are more likely to develop CLBP after depression (OR=1.58, 95% CI [1.17, 2.14], adj.p = 4.5x10$^{-6}$),

**Table 2. Enrichment for individual pain conditions observed in females and racial minorities.**

| Disease | Cases | Sex | Race | | | | |
|---|---|---|---|---|---|---|---|
| | | Female | Asian | Black or African American | Middle Eastern or North African | More than One Population | American Indian or Alaska Native |
| CFS | 3,563 | 1.86*** | ns | 0.39*** | ns | ns | ns |
| CLBP | 51,480 | 1.16*** | 0.82* | 1.24*** | ns | 1.36*** | ns |
| CTTH | 764 | 2.10*** | ns | ns | ns | ns | ns |
| FM | 6,469 | 9.99*** | 0.35*** | 0.59*** | ns | ns | ns |
| IBS | 8,834 | 2.84*** | 0.36*** | 0.41*** | ns | ns | ns |
| MHA | 23,633 | 3.31*** | 0.38*** | 0.61*** | ns | ns | ns |
| TMD | 2,381 | 3.18*** | ns | 0.61** | ns | ns | ns |
| UCPPS | 947 | ns | ns | ns | ns | ns | ns |
| Anxiety | 62,153 | 1.50*** | 0.37*** | 0.54*** | 0.64** | ns | 0.75* |
| Bipolar | 8,744 | ns | 0.31*** | 0.83** | 0.36* | 1.31* | ns |
| Depression | 60,001 | 1.48*** | 0.41*** | 0.68*** | 0.70* | ns | ns |
| Schizophrenia | 2,247 | 0.41*** | ns | 2.61*** | ns | 1.89* | ns |

*Note*: OR: odds ratio. ns: not significant after Bonferroni correction, adj.p > 0.05, *: $10^{-3} \leq$ adj.p < 0.05, **: $10^{-6} \leq$ adj.p < $10^{-3}$, ***: adj.p < $10^{-6}$. The reference group for sex is male, and the reference group for race is White. Since no significant racial or ethnic disparity factors have been identified for endometriosis and vulvodynia—female-specific disorders—they are not shown in the table. Of note, AoU includes a racial minority of Middle Eastern or North African for descendants of the region.

anxiety (OR 1.50, 95% CI [1.12, 2.00], adj.p = $4.6 \times 10^{-5}$), or MHA (OR=1.44, 95% [1.02, 2.05], adj.p = $1.4 \times 10^{-2}$). Persons with the lowest household income are also more likely to develop MHA after CLBP (OR=2.05, 95% CI [1.40, 3.01], adj.p = $1.7 \times 10^{-9}$), IBS (OR=1.99, 95% CI [1.00, 3.95], adj.p = $2.6 \times 10^{-2}$), and depression (OR=2.15, 95% CI [1.42, 3.23], adj.p = $2.4 \times 10^{-9}$). In addition, they were more likely to develop IBS after MHA (OR=2.25, 95% CI [1.22, 4.16], adj.p = $1.6 \times 10^{-4}$) and anxiety (OR=1.84, 95% CI [1.01, 3.34], adj.p = $2.0 \times 10^{-2}$). Finally, this group is more likely to develop anxiety (OR=1.31, 95% CI [1.02, 1.68], adj.p = $8.5 \times 10^{-3}$) and depression (OR=1.37, 95% CI [1.02, 1.83], adj.p = $9.6 \times 10^{-3}$) after CLBP, and develop anxiety after MHA (OR=1.38, 95% CI [1.01, 1.88], adj.p = $1.6 \times 10^{-2}$) compared to the reference group.

## Materials and methods

### Dataset and phenotype definition

We used the AoURP [18] version 8, released in February 2025, which involved the electronic health records (EHR) of over 633,000 participants. Only individuals sharing EHR data were included, and those whose data showed apparent quality or power issues were excluded, such as problematic dates, ages (over 110 years old in any record), less than two clinical visits, and disagreement between reported sex and gender. Variables studied include onset of each COPC and mental condition, sex at birth, age, race, ethnicity, household income level, and region of state residence (e.g., northeast). Sex at birth, race, and ethnicity were self-reported during the enrollment of AoU.

The determination of COPC/mental cases worked as follows:

(1) Identifying episodes of COPC and mental conditions: We cataloged COPC and mental health conditions using the ICD-10 [23] and ICD-9 [24] codes recognized by the research community. Utilizing terms from the literature and descriptions, we linked the conditions to the corresponding AoURP standard concept ids, which served as the primary inclusion criteria.

(2) Establishing chronicity: A condition was deemed chronic if it was diagnosed with at least two separate episodes, a minimum of 90 days apart, aligning with standards accepted by the pain research community [24].

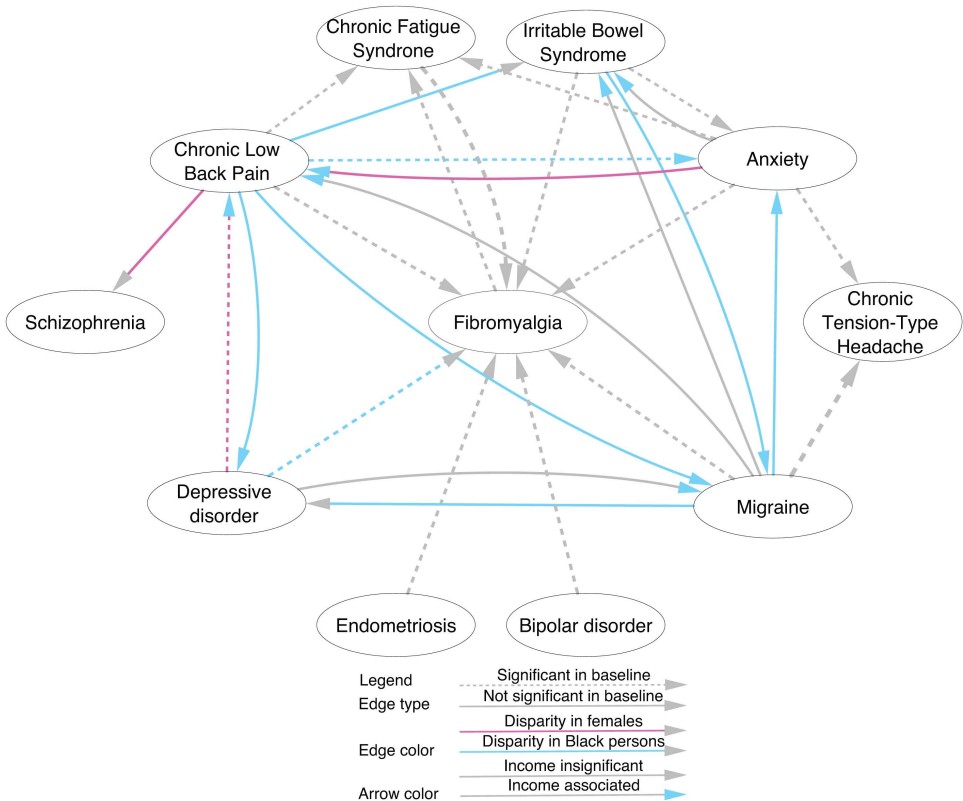

**Fig 1. The overrepresentation of specific pairwise relations between COPC and mental conditions, comparing females, racial minority persons, and persons with low income to the reference sociodemographic groups.** Each node represents either a specific COPC condition or a mental disorder. For sociodemographic reference in the statistical analysis, male is the reference sex, White persons make the reference race, non-Hispanic White is the reference ethnicity, and the highest income stratum is the reference income level. Each arrowed edge depicts a significant temporally ordered disease pair supported by either the reference demographics (dashed edges) or the alternative demographics (solid edges). Edge colors indicate significant (Bonferroni-correlated p-value<0.05) association with specific demographics (gray for White, blue for Black or African American, and purple for females), while arrow colors indicate significant overrepresentation of the lowest household income levels in contrast to the highest level (gray for none and blue for significant).

(3) Determining the earliest onset: To establish each patient's order of distinct conditions, we identified the earliest onset for each condition based on diagnosis dates. Additionally, we required that the onset of each condition of a patient occur at least one year (clearance period) after his/her first EHR record, to minimize the noise from using incorrect onsets. We excluded the persons who had more than one COPC or mental condition within the clearance period, for which the order of the condition was difficult to establish.

## Cohort creation process

We built a separate cohort for each demographic factor to assess its influence on developing a COPC/mental disease or progressing to another COPC. The process of cohort selection for each analysis proceeded as follows:

(1) Building a cohort for each demographic variable that developed a COPC/mental condition: Each group (e.g., African American) of a demographic variable (e.g., race) was compared to a reference group (e.g., White). We constructed a cohort of participants based on the assessed or the reference group. The same number of participants were selected for each of the two groups, further matched by other variables (not being assessed), such as sex at birth, birth year

(categorized in decade intervals), and race. We conducted random sampling without replacement from the group that had more samples (usually the reference group, e.g., White), ensuring an equivalent number of participants from the larger group for each demographic combination. Infrequently, when there were not enough samples to achieve a complete demographic match for a particular demographic feature combination, we opted for partial matching. The precedence for matching was sex, followed by birth year group, and then race. We did not include gender, which is a broader social construct concept, due to its high correlation with biological sex and insufficient data to reach the required power.

(2) Building a cohort for each disease progression with at least 21 occurrences (per AoU policy): Starting from any COPC/mental condition, we divided the participants into two sub-cohorts, those with and without the start condition. We then matched the patients with the starting COPC/mental condition against those without the condition, by sex, age interval, and race, similar to the procedure above, but matching each case up to a sufficient number of controls to get the maximum power. The case:control ratios ranging from 1:1–1:10 were explored. Then, we studied the difference in the onset of other conditions after the start of the COPC/mental condition between the two sub-cohorts of participants. Lastly, we assessed the differential influence of a sociodemographic group as compared to their reference group, with regard to the disease progression patterns within this specific cohort.

## Statistical analysis

After building the cohort for each comparison, we assessed the increased risk of developing the first or progressing to another condition. Each COPC/mental condition other than the start condition (for the progression study only) served as an outcome variable. We employed a logistic regression model to assess the statistical significance of each sociodemographic group, specifically the odds ratio of developing or progressing to the outcome condition, relative to the reference group. In the COPC onset analysis, odds ratios were derived from the coefficient of a sociodemographic variable for risk of outcome, while in the progression analysis, the odds ratios were from the interaction term between the first condition and the sociodemographic variable. An odds ratio exceeding 1 with statistical significance would suggest disparity associated with the sociodemographic factor.

The demographic variables assessed include sex, race, and ethnicity, for the known relevance in chronic pain [7,12–14,25–27]. Additional covariates were incorporated including birth year, household income level, and region of state of residence. Birth year was treated as a numerical variable, whereas sex, race, ethnicity, and income level were handled as categorical variables. Income levels—stratified as below $25k, $25-$50k, $50-$100k, $100-$150k, $150-$200k, and over $200k—served as a proxy for socioeconomic status. Missing income data (Skip or Prefer not to answer in questionnaire response) was treated as a separate category.

To address potential multiple comparison errors, we applied the Bonferroni correction for adjusting confidence intervals (CI) for odds ratios and p-values (adj.p).

## Ethics statement

The study utilized de-identified data from the All of Us Research Program. The University of Arizona Institutional Review Board (IRB) reviewed the study titled "Identify the heterogeneity and commonality of chronic overlapping pain conditions (COPCs) through phenotypic and genomic perspectives" (IRB Submission ID: STUDY00001119) and determined that it does not qualify as human research, as defined by DHHS and FDA regulations. Consequently, IRB review and approval by this organization is not required.

## Discussion

Our results align with known disparities in developing COPC and their comorbidities, e.g., the elevated risk for females to develop all COPCs [28,29], except UCPPS, bipolar disorder, and schizophrenia [30]. Possible explanations may involve

biological differences between sexes [13], such as hormone factors [28], sensitization [17], and inflammatory responses [25]. Furthermore, this study provides a distinct contribution by clarifying disparities associated with specific progression patterns in COPCs and mental conditions [31]. For instance, our findings indicate that females are at a significantly heightened risk for developing chronic low back pain following an anxiety or depression diagnosis. Although the impact of psychological conditions on chronic low back pain has been widely studied [32–36], the influence and magnitude of sex differences in the progression from a mental disorder to CLBP remain inconclusive. These findings offer new insights into this issue.

The results indicated an increased risk of developing chronic low back pain (CLBP) in Black or African Americans [17], who may also face a higher risk for subsequent COPCs (e.g., irritable syndrome) and mental conditions (e.g., anxiety and depression). Although CLBP is more prevalent among African Americans, its differential influence on other COPC pains or mental conditions has been rarely reported in Black Americans, particularly by longitudinal and prospective studies [37]. A few studies suggested Black population with chronic pains reported more depressive symptoms [38]. This may be related to the racial differences in pain sensitivity and perception [26,39–41], psychological response to chronic pain [42], stress levels [38], and allostatic load [43,44]. Of note, socioeconomic confounders cannot be ruled out in the context of health-care access and utilization [31,45,46], which would influence the management and therefore containment of disease progression.

While sociodemographic disparities in the development of individual COPC conditions have been frequently reported, studies investigating such disparities in their progression are rare. Not only are multiple COPCs associated with more complex outcomes [47], but knowledge gained about the differential progression should be translated into optimization of chronic pain management, cross-specialty coordination, and creation of multi-faceted support systems customized for every sociodemographic group, including their intersections.

Not all observed progressions were statistically significant across demographic variables (e.g., sex) for certain pairs of COPCs. This outcome was likely due to the underpowered nature of certain cohort groups, limiting the ability to detect subtle differences in COPC progression. The limitation in statistical power may stem from the sample sizes for progression between some COPC pairs [48] (e.g., between CTTH and TMD) or in certain strata, particularly when the data was segmented by multiple dimensions such as age, sex, race, and income level. The observation may also be attributed to the modeling of socioeconomic effects in the risk of progression since many associations between the COPC to mental conditions could be confounded by socioeconomic factors. For instance, a lack of sufficient medical care may exacerbate a condition [49]. Finally, sex may contribute more to the development of the first condition, while not contributing significantly to the progression [50–52], which warrants further verification [53]. In addition, we did not identify any disparity for the Hispanic populations, neither in baseline nor in progression. Further research is needed to reproduce and explain the findings.

The study and results rely heavily on electronic health records, which are known to have potential biases and noises. The All of Us Research Program is known to be enriched with underrepresented populations and diverse health conditions, which could potentially affect the effect size of the results.

Future research should be able to build on the discovered correlations, exploring the causes and effects of these correlations. Our progression analysis represents a promising approach to health disparity research, especially as more data becomes available. For example, there is limited data for studying chronic pain disparities experienced by American Indians [54] or intersex persons [13]. Such research is crucial for developing effective pain management strategies to address the needs of specific populations [55].

In conclusion, the study demonstrated a promising approach to systematic discovery of the differential chronic pain progression patterns across sociodemographic groups. Our results not only reinforced existing evidence, but also uncovered lesser-known associations, such as complications of increased chronic low back pain in Black or African American populations. By recognizing the confluence of sociodemographic factors, we anticipate such data-driven research will enable more contextualized strategies in caring for our diverse populations affected by chronic pains and encourage future prospective studies to build stronger evidence.

## Supporting information

**S1 File. Logistic regression test results for COPC and mental disorder pairs with more than 20 cases in the All of Us Research Program (Version 8).** This supplementary file details the logistic regression outcomes for pairs of COPC and mental disorders, each involving over 20 cases. The table displays test results for outcome conditions (in the 'right' column) for the baseline group—Non-Hispanic White males with the highest income level (annual household income of $200,000 or more)—as well as the effects of various non-reference demographic groups and income levels. Additionally, interaction terms, which capture differential progression patterns from a precedent condition (in the 'left' column), are indicated in columns prefixed with "left.".
(TXT)

## Acknowledgments

We gratefully acknowledge All of Us participants for their contributions, without whom this research would not have been possible. We also thank the National Institutes of Health's All of Us Research Program for making available the data and analytics platform.

## Author contributions

**Conceptualization:** Edwin Baldwin, Jin Zhou, W. Michael Hooten, Jungwei W. Fan, Haiquan Li.

**Formal analysis:** Edwin Baldwin, Jin Zhou, Haiquan Li.

**Funding acquisition:** Jin Zhou, W. Michael Hooten, Haiquan Li.

**Investigation:** W. Michael Hooten, Jungwei W. Fan, Haiquan Li.

**Methodology:** Edwin Baldwin, Wenting Luo, Jungwei W. Fan.

**Project administration:** Jungwei W. Fan, Haiquan Li.

**Software:** Jungwei W. Fan.

**Supervision:** Jin Zhou, W. Michael Hooten, Jungwei W. Fan, Haiquan Li.

**Validation:** Edwin Baldwin, Wenting Luo, Haiquan Li.

**Visualization:** Edwin Baldwin.

**Writing – original draft:** Edwin Baldwin.

**Writing – review & editing:** Edwin Baldwin, Jin Zhou, W. Michael Hooten, Jungwei W. Fan, Haiquan Li.

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
