## [Decision Letter · Decision Letter 0]

PDIG-D-24-00489Incidence and Progression Patterns of Chronic Pain Disparities Revealed by Analyzing the All of Us Research Program DataPLOS Digital Health Dear Dr. Li, Thank you for submitting your manuscript to PLOS Digital Health. After careful consideration, we feel that it has merit but does not fully meet PLOS Digital Health's publication criteria as it currently stands. Therefore, we invite you to submit a revised version of the manuscript that addresses the points raised during the review process. Please submit your revised manuscript within 60 days Feb 16 2025 11:59PM. If you will need more time than this to complete your revisions, please reply to this message or contact the journal office at digitalhealth@plos.org. Please include the following items when submitting your revised manuscript:* A rebuttal letter that responds to each point raised by the editor and reviewer(s). You should upload this letter as a separate file labeled 'Response to Reviewers '. This file does not need to include responses to any formatting updates and technical items listed in the 'Journal Requirements' section below.* A marked-up copy of your manuscript that highlights changes made to the original version. You should upload this as a separate file labeled 'Revised Manuscript with Track Changes '.* An unmarked version of your revised paper without tracked changes. You should upload this as a separate file labeled 'Manuscript '. If you would like to make changes to your financial disclosure, competing interests statement, or data availability statement, please make these updates within the submission form at the time of resubmission. Guidelines for resubmitting your figure files are available below the reviewer comments at the end of this letter. We look forward to receiving your revised manuscript. Kind regards, Miguel E Rentería, PhDAcademic EditorPLOS Digital Health Leo Anthony CeliEditor-in-ChiefPLOS Digital Healthorcid.org/0000-0001-6712-6626 **Journal Requirements:**

2. We have amended your Competing Interest statement to comply with journal style. We kindly ask that you double check the statement and let us know if anything is incorrect.

 **Additional Editor Comments (if provided):** Dear A/Prof Li,

Thank you for submitting your manuscript, "Incidence and Progression Patterns of Chronic Pain Disparities Revealed by Analyzing the All of Us Research Program Data," to PLOS Digital Health. After careful consideration of the reviewers' comments, I would like to invite you to submit a revised version of your manuscript. The reviewers have highlighted the strengths of your study, including its focus on underrepresented populations and the innovative use of the All of Us dataset, but have also identified areas where improvements are needed.

You can find the reviewers comments below.**Reviewers' Comments:** Reviewer's Responses to Questions

**Comments to the Author**

1. Does this manuscript meet PLOS Digital Health’s publication criteria ? Is the manuscript technically sound, and do the data support the conclusions? The manuscript must describe methodologically and ethically rigorous research with conclusions that are appropriately drawn based on the data presented.

Reviewer #1: Partly

Reviewer #2: Yes

2. Has the statistical analysis been performed appropriately and rigorously?

Reviewer #1: No

Reviewer #2: Yes

3. Have the authors made all data underlying the findings in their manuscript fully available (please refer to the Data Availability Statement at the start of the manuscript PDF file)?

Reviewer #1: No

Reviewer #2: No

4. Is the manuscript presented in an intelligible fashion and written in standard English?

Reviewer #1: No

Reviewer #2: Yes

5. Review Comments to the Author

Reviewer #1: This paper by Edwin Baldwin et al., titled "Incidence and Progression Patterns of Chronic Pain Disparities Revealed by Analyzing the All of Us Research Program Data” explores a valuable contribution to understanding the demographic disparities in the progression of chronic overlapping pain conditions (COPC) and associated mental health conditions.

One of the strengths of the study is the use of the All of Us Research Program dataset, which provides a large and diverse cohort that allows for a more nuanced examination of how sex, race, and ethnicity impact the progression of chronic pain and comorbid mental health disorders.

There are a few points that the authors should address to improve the paper:

1. Introduction

The introduction is well-written and sets the stage by addressing the broader context of disparities in chronic pain and mental health. The focus on underrepresented populations is both timely and highly relevant. However, this section could be enhanced by a deeper explanation of why certain demographic groups experience more severe progression of COPCs. Adding citations on factors like unique pain mechanisms, social determinants, or healthcare barriers specific to these populations could offer valuable context and strengthen the argument.

On the other hand, the concept of COPCs could be more clearly defined, as it currently lacks sufficient detail.

Additionally, it would be helpful if the authors elaborated on the rationale behind selecting specific COPCs and mental health conditions.

2. Methods

- The use of logistic regression for analysing COPC progression is appropriate, yet further details on the model's covariates would improve the reproducibility of your findings.

- The study’s use of the All of Us dataset offers a robust data source. However, addressing potential biases related to electronic health records (EHR) is essential—especially in underrepresented populations. Highlighting any limitations in the EHR’s ability to capture pain-related outcomes or disparities could help contextualize the findings.

- While the exclusion criteria for EHR quality issues are mentioned, additional details on the specific types of data quality issues (e.g., missing data, inconsistencies).

- While “disparity factors” are mentioned, the rationale for selecting these specific factors could be further developed. Adding references to supporting literature on the relevance of these factors for COPC and mental health disparities.

- Providing context on why certain reference levels (e.g., “White” for race) were chosen would clarify the model’s baseline comparisons and reinforce the methodological rigor.

- The criteria for identifying disease progression between multiple COPC or mental health conditions could be more explicit. Specifying whether progression is tracked between distinct conditions (e.g., from anxiety to depression) or within subtypes.

- Details on how income levels were categorized and whether adjustments were made for regional cost-of-living differences.

4. Results:

- The table 1 includes “Other or Unknown” categories for both sex and race, but there is no definition for what constitutes “Other” or “Unknown.” Since sex is generally defined as a biological classification based on physical and physiological traits (e.g., chromosomes, hormone levels, and anatomy), could you provide data to support this classification? Alternatively, if the study uses this term in a broader, social context—referring to gender (a social construct encompassing roles, behaviors, and identities)—this distinction should be made clear.

- The label “COPC Incidence” may be misleading, as the data appears to show the prevalence of each condition within the study cohort, rather than the incidence (rate of new cases over time). Adjusting this label to reflect “prevalence” would prevent misinterpretation of the data.

- Please explain the criteria participants used for self-identification or how race and ethnicity data were gathered, as variations in options could lead to inconsistency in reporting. The “Other or Mixed” racial category comprises a relatively large portion (3.3%) but lacks specificity. Clarifying which racial or ethnic identities are included. Additionally, please specify if “Other or Mixed” refers to a blend of ethnic backgrounds or individuals identifying with multiple racial identities.

- The term “non-White races” is somewhat vague (Table 2). Using more specific language, such as “racial and ethnic minorities,” or explicitly listing racial groups (e.g., “Black or African American, Asian, Native Hawaiian or Other Pacific Islander”).

- The table includes categories such as “Middle Eastern or North African,” “More than One Population,” and others. If these categories were self-reported, providing definitions for each at the beginning of the study would be helpful, as some may overlap or be interpreted differently.

- To enhance readability, please provide a legend for clarify “ns” as “non-significant, p > 0.05.” ”insignificant” does not meet the acronymic nomination criteria.

- While endometriosis (ENDO) and vulvodynia (VVD) are appropriately noted as “exclusive” to females, a note explaining these as conditions specific to individuals with female reproductive anatomy would provide helpful context for those less familiar with these terms. Consider not include in the Table this information.

The terms “non-White races” and “reference demographics” (White, non-Hispanic, male) could benefit from more precise language.

- Clarify the “solid edges with hollow triangle arrows” by noting their significance, such as “solid edges indicate a strong association,” which would enhance visual understanding. While color coding by demographic group is effective, providing a legend specifying which color represents each group (e.g., “Black arrows for Black individuals, orange for Hispanic individuals”).

- To improve comprehension, a brief introductory explanation of the node and connection symbols would be helpful before diving into the detailed analysis. This will make the visual representation more accessible to readers unfamiliar with these symbols.

- Including confidence intervals alongside the odds ratios.

4. Discussion:

The discussion would benefit from further comparison with previous studies that have examined disparities in single COPCs. While the authors highlight the novelty of their approach in studying progression disparities, further contextualisation with similar research could help underscore the impact of this study.

A section on potential mechanisms driving the observed disparities would add depth to the discussion. For instance, expanding on biological, social, and psychological explanations for increased pain susceptibility or mental health conditions in certain populations (such as higher inflammation rates in females or the role of socioeconomic stressors in marginalised groups) could provide valuable insights.

Reviewer #2: Comments:

This study investigated the disparities in progressions in developing COPCs and mental conditions. The findings of this study added an important contribution to understand the interactions of COPCs and mental conditions among underrepresented populations and develop effective COPCs management strategies.

Abstract:

1. Lines 24-26. Why reported subsequent depression after irritable bowel syndrome as an example in abstract? Because the disparities in this progression pathway is the most significant one? Or any other reasons? As a reader, I would also wonder what the nine progression pathways are. I understand that the word count is limited for the abstract, any possible ways to mentioned two or three out of the nine?

Introduction

Major comment:

2. The second half of the introduction are mostly about the study’s objectives but rarely study innovation. Lines 51-55, you mentioned that previous evidence in disparities in the progressions of COPCs are limited. Why you think this is an important question? I see that you only mentioned in lines 58-59 that they “were of special interest for their well-documented interactions with COPCs”. Can you add more justification for the study in the introduction?

Minor comment:

3. Lines 44-45, “including anxiety, depression, bipolar disorder, and schizophrenia” are the four comorbid mental health conditions mentioned in line 57, right? If yes, would add “four” before “mental health conditions” in line 44, as. You did for the ten chronic pains to help readers quickly link them together.

Results:

Minor comments:

4. Line 67, what is AoURP v7 dataset? Version 7 or specific survey year? Please clarify.

5. Table 1, any missing cases? How are they excluded?

6. Lines 72-84, and the entire results section, sometime p value is exactly reported (e.g. CLBP, p<9.18*10-15), sometime p value is reported in a format of p<10-10 (line 83). Please keep using a consistent format. Suggest the later way.

7. Fig.1 This is a nice figure summarizing the findings of the interactions between COPC and mental conditions. Suggest adding corner labels/legends to clarify black arrows indicates significant disparities among Black persons, orange arrows indicates Hispanic/Latino, etc. I see they are described as texts in lines 95-98. Corner labels/legends will complete the figure so that readers can quickly catch the key findings.

Methods:

Minor comments:

8. 127, can you clearly define/list your reference group. I read that it is male for sex, Hispanic for ethnicity, White for race in results section. But I think you need define them clearly in methods first, and why they are selected as the reference group?

6. PLOS authors have the option to publish the peer review history of their article (what does this mean? ). If published, this will include your full peer review and any attached files.

**Do you want your identity to be public for this peer review?** For information about this choice, including consent withdrawal, please see our Privacy Policy .

Reviewer #1: No

Reviewer #2: No

---

## [Decision Letter · Decision Letter 1]

Sociodemographically Differential Patterns of Chronic Pain Progression Revealed by Analyzing the All of Us Research Program Data

PDIG-D-24-00489R1

Dear Li,

We are pleased to inform you that your manuscript 'Sociodemographically Differential Patterns of Chronic Pain Progression Revealed by Analyzing the All of Us Research Program Data' has been provisionally accepted for publication in PLOS Digital Health.

Best regards,

Miguel E Rentería, PhD

Academic Editor

PLOS Digital Health

**Additional Editor Comments (if provided):**

**Reviewer Comments (if any, and for reference):**

Reviewer's Responses to Questions

**Comments to the Author**

1. If the authors have adequately addressed your comments raised in a previous round of review and you feel that this manuscript is now acceptable for publication, you may indicate that here to bypass the “Comments to the Author” section, enter your conflict of interest statement in the “Confidential to Editor” section, and submit your "Accept" recommendation.

Reviewer #1: All comments have been addressed

Reviewer #2: All comments have been addressed

2. Does this manuscript meet PLOS Digital Health’s publication criteria ? Is the manuscript technically sound, and do the data support the conclusions? The manuscript must describe methodologically and ethically rigorous research with conclusions that are appropriately drawn based on the data presented.

Reviewer #1: Yes

Reviewer #2: Yes

3. Has the statistical analysis been performed appropriately and rigorously?

Reviewer #1: Yes

Reviewer #2: Yes

4. Have the authors made all data underlying the findings in their manuscript fully available (please refer to the Data Availability Statement at the start of the manuscript PDF file)?

Reviewer #1: Yes

Reviewer #2: Yes

5. Is the manuscript presented in an intelligible fashion and written in standard English?

Reviewer #1: Yes

Reviewer #2: Yes

6. Review Comments to the Author

Reviewer #1: (No Response)

Reviewer #2: I appreciate the authors' thorough and thoughtful responses to the reviewer comments. I find that the revisions and clarifications have substantially improved the manuscript. I think the authors' responses are valid and enhance the overall quality of the work.

7. PLOS authors have the option to publish the peer review history of their article (what does this mean? ). If published, this will include your full peer review and any attached files.

**Do you want your identity to be public for this peer review?** For information about this choice, including consent withdrawal, please see our Privacy Policy .

Reviewer #1: No

Reviewer #2: No
